# Proteases and Their Potential Role as Biomarkers and Drug Targets in Dry Eye Disease and Ocular Surface Dysfunction

**DOI:** 10.3390/ijms23179795

**Published:** 2022-08-29

**Authors:** Alba Ramos-Llorca, Camilla Scarpellini, Koen Augustyns

**Affiliations:** Laboratory of Medicinal Chemistry, Department of Pharmaceutical Sciences, Faculty of Pharmaceutical, Biomedical and Veterinary Sciences, Campus Drie Eiken, University of Antwerp, Universiteitsplein 1, B-2160 Antwerp, Belgium

**Keywords:** dry eye, inflammation, ocular surface, proteases, protease-activated receptors, protease inhibitor

## Abstract

Dry eye disease (DED) is a multifactorial disorder that leads to ocular discomfort, visual disturbance, and tear film instability. DED is accompanied by an increase in tear osmolarity and ocular surface inflammation. The diagnosis and treatment of DED still present significant challenges. Therefore, novel biomarkers and treatments are of great interest. Proteases are present in different tissues on the ocular surface. In a healthy eye, proteases are highly regulated. However, dysregulation occurs in various pathologies, including DED. With this review, we provide an overview of the implications of different families of proteases in the development and severity of DED, along with studies involving protease inhibitors as potential therapeutic tools. Even though further research is needed, this review aims to give suggestions for identifying novel biomarkers and developing new protease inhibitors.

## 1. Introduction

Dry eye disease (DED), defined by the Tear Film and Ocular Society (TFOS) in the Dry Eye Workshop II (DEWS) report, is a multifactorial disease of the ocular surface defined by the disruption of homeostasis of the tear film. The ocular signs accompanying DED are tear film instability and hyperosmolarity, ocular surface inflammation and damage, and neurosensory abnormalities [1]. Consequently, some symptoms patients describe are dryness, itching, redness, visual disturbance, and ocular fatigue. The impact of DED on the quality of life increases with the disease severity [2]. In general, DED affects the ability to perform daily activities and work, and in more severe cases, it can instigate mood alterations and depression [3]. The prevalence of DED ranges from 5% to 50%, depending on the studied population [4]. However, prevalence studies can differ due to a lack of heterogeneity in the description of DED and whether the study is based on the symptoms or the signs of the patients [5]. Risk factors for DED include sex and race [6]. Furthermore, the constant use of screens, wear of contact lenses, environmental conditions, and use of medication are also considered risk factors [7]. Thus, with the current general population lifestyle, the prevalence is expected to increase in the following years. For instance, during the recent COVID-19 pandemic, a rise in patients with DED symptoms was described. This increase is correlated to wearing a face mask, also known as mask-associated dry eye. The misplacement of the face mask potentially displaces the air around the eyes and increases the evaporation of tears [8,9].

DED patients can be broadly divided into evaporative dry eye (EDE) and aqueous deficient dry eye (ADDE) [10]. They are not exclusive; thus, patients can present characteristics of the two simultaneously. EDE is associated with the dysregulation of the lipid layer of the tear film. This leads to excessive evaporation of the aqueous layer, causing hyperosmolarity and inflammation of the ocular surface [11]. This DED type is commonly associated with meibomian gland dysfunction (MGD). The meibomian glands are responsible for segregating lipids towards the eyelid margin to form the tear film lipid layer. In patients with MGD, the quantity and quality of the lipids are decreased [12]. Thus, MGD can lead to DED. On the contrary, ADDE is correlated with reduced aqueous production by the lacrimal system or the accessory glands [13]. ADDE, in itself, can be divided into Sjögren syndrome dry eye (SSDE) and non-Sjögren syndrome dry eye (NSSDE) [1]. Sjögren syndrome is an autoimmune disease affecting the exocrine glands, specifically the salivary and lacrimal glands, resulting in dryness of mucosal surfaces [14]. The DED pathophysiology can be described by the vicious circle of dry eye [15]. The ocular surface disruption leads to osmotic stress. Then, hyperosmolarity initiates stress-related signaling pathways and the release of inflammatory cells and cytokines [16]. Among the activated signaling pathways, there are the nuclear factor kappa beta (NFκB), the mitogen-activated protein kinase (MAPK), and the c-Jun N-terminal kinase (JNK) [17].

The current diagnosis of DED is based on questionnaires handed to patients. The Ocular Surface Disease Index (OSDI) is the most widespread and widely used, which assesses symptoms, environmental triggers, and the impact on quality of life. Other tests widely performed by clinicians include tear film breakup time, fluorescein breakup time, and the Schirmer test [18]. Tear film biomarkers for DED are proinflammatory cytokines and chemokines, such as interleukin-6 (IL-6), tumor necrosis factor alpha (TNF-α), IL-1β, interferon gamma (INF-γ), and IL-8 [19]. Other biomarkers include matrix metalloprotease-9 (MMP-9) and vascular endothelial growth factor. A limitation of biomarker identification is the variability among different methods of collection and instruments [20]. 

The most common treatment is artificial tear substituents. These bring momentary relief by diluting inflammatory markers. However, artificial tears do not have an anti-inflammatory effect; therefore, they do not tackle the main trigger of DED [21,22]. Cyclosporin A (CyA) was the first approved drug by the Food and Drug Administration (FDA, USA) to treat specifically DED. CyA is an immunomodulatory drug that reduces many inflammatory markers. Unfortunately, many patients do not respond to CyA or have significant side effects [23]. Xiidra, a competitive antagonist of lymphocyte function-associated antigen-1 (LFA-1), was approved in 2016. It takes up to 3 months to reduce the symptoms, and many patients experience side effects [24]. Finally, topical corticosteroids, such as Eysuvis, approved in 2020, can only be used for short periods due to steroid side effects [25]. 

To date, the diagnosis and treatment of DED present many challenges. Thus, there is a significant interest in identifying new potential biomarkers and developing new therapies. 

In a healthy eye, proteases and endogenous protease inhibitors are in equilibrium in the tear film. On the contrary, the imbalance of their levels can induce different pathologies [26]. Proteases destroy peptides or proteins by cleaving peptide bonds through hydrolysis. [27]. Their function is essential in living organisms [28]. Proteases are encoded by about 2% to 4% of the total genes in all kinds of organisms [29]. The MEROPS database (12.0 release), an information source about all peptidases and their inhibitors, includes more than 5000 proteases [30]. Based on their catalytic mechanism, they are divided into seven families: aspartic, cysteine, glutamic, metallo-, asparagine, serine, and threonine proteases [31]. Metallo- and serine proteases are the most abundant families [29].

Proteolytic enzymes have raised a great interest in biomedical research due to their involvement in many physiological processes. Proteases are relevant in regulating many diverse biological processes [32]. For example, they modulate protein–protein interactions, cell division and replication, wound repair, blood coagulation, digestion, immunity, and inflammation [33,34]. Due to their role in pivotal biological functions, the malfunction or alteration of their expression is correlated with pathological conditions, such as cancer, inflammatory, cardiovascular, and neurodegenerative disorders [28]. Proteases can also act in signaling pathways by activating protease-activated receptors (PARs) [35]. 

Recently, studies involving proteases in the ocular surface have focused on matrix metallo-, serine, and cysteine proteases [36]. In particular, for DED, the role of MMP-9 has been widely studied [19]. Even though the role of other proteases is less well established, there is evidence that serine and cysteine proteases also play a role in the immunity and inflammation of DED [37,38].

This review assesses the role of proteases and PARs in ocular surface dysfunction and concentrates more specifically on their potential role as biomarkers and therapeutic targets in DED. Protease inhibitors with the potential to mitigate symptoms of DED are also discussed.

## 2. Proteases and Dry Eye Disease

Proteases play pivotal roles in inflammation and are considered therapeutic targets and biomarkers for different pathologies. Different reviews can be found on this subject [28,39,40]. The correlation between DED and inflammation and the involvement of proteases in inflammatory events show the potential for proteases to become new drug targets or biomarkers for DED. The following section describes the different families of proteases linked to ocular surface disorders. We summarize their role in inflammation and immunity and their involvement in various pathologies, specifically in DED (Figure 1).

### 2.1. Matrix Metalloproteases

MMPs are calcium-dependent proteases that contain a zinc atom. They are responsible for tissue remodeling and extracellular matrix degradation [41,42]. MMPs can be found as secreted or transmembrane proteases. However, they are synthesized as inactive zymogens and activated by different proteases through a process called cysteine switch [43]. Under healthy physiological conditions, the activity of MMPs is further regulated by their endogenous inhibitors, known as tissue inhibitors of MMPs (TIMPs). The equilibrium between MMPs and TIMPs is crucial to control their activity [44]. 

MMPs have been shown to affect inflammatory pathways and immune responses in different pathologies, including DED. The involvement of MMPs in DED has been extensively studied. For example, in an experimental dry eye model, the expression of several MMPs (MMP-1, 10, 3, and 9) increased under desiccating stress conditions [45]. In many reports, the critical role of MMP-9 in response to hyperosmolar stress in DED is described [46].

The expression of MMP-9 has been studied in several experimental DED animal models. For instance, DED was induced in C57BL6 mice by scopolamine hydrobromide and exposure to airflow [47], or extra orbital lacrimal gland removal [48]; Balb/C mice were treated with benzalkonium chloride (BAC) [48]; or in New Zealand white rabbits, DED was induced by concanavalin A [49]. In all these experimental settings, the expression of MMP-9 was enhanced, tear production was reduced, and corneal epithelial damage could be measured [47,48,49]. Interestingly, the induction of DED to MMP-9 knockout mice improved resistance to corneal epithelial barrier damage [50].

Clinical studies report the use of MMP-9 as a DED biomarker due to a direct correlation between its upregulation and the development and severity of the disease [51,52,53]. Moreover, Kook et al., in a study involving 63 SSDE patients, determined that the concentration of tear MMP-9 combined with tear osmolarity may help determine the severity of SSDE [54]. An immunoassay called InflammaDry is commonly used clinically to measure MMP-9 levels and discriminate between DED and non-DED patients [55]. 

### 2.2. Serine Proteases

Serine proteases are named after the presence of a nucleophilic serine in the active site. The active site is composed of a catalytic triad, which is conserved among most of the proteases of this family [56]. Serine proteases can be further divided by their substrate specificity in chymotrypsin-, trypsin- and elastase-like serine proteases [57]. They are an abundant proteolytic enzyme family that plays pivotal roles in critical physiological processes. Their upregulation leads to different malfunctions and pathologies [58]. Most of the serine proteases are secreted to the extracellular milieu. However, a reduced number can be found intracellularly or membrane-bound [59]. In a healthy environment, endogenous inhibitors, known as SERPINS, regulate serine protease activity [60]. 

Some serine proteases are directly involved with the immune system and secreted by immune cells. For instance, neutrophil proteases are expressed in granulocytes, granzymes in lymphocytes, and tryptase and chymase in mast cells [61]. These promote pro-inflammatory cytokine expression and MMP-9 activation and impact the degradation of extracellular matrix components and the loss of epithelial barrier function. There is evidence of their contribution to several diseases, such as arthritis, asthma, irritable bowel syndrome, and multiple sclerosis [62,63]. Serine proteases and their endogenous inhibitors have been found in the tear fluid on the ocular surface [64]. While studies that focus on the presence of serine proteases in DED patients are lacking, the dysregulation of expression of related immune cells hypothesizes that serine proteases might be secreted upon degranulation to the extracellular milieu. 

Neutrophils are an innate immune cell type known to act upon infection or injury. Additionally, they play roles in adaptive immunity by interacting with T and B cells [65]. They play a role in other autoimmune diseases, such as rheumatoid arthritis [66]. Upon degranulation, serine proteases (cathepsin G, neutrophil elastase, and proteinase-3) are released [67]. Evidence suggests that neutrophils play a role in DED. Their presence has been described in several studies [68,69]. Recently, Postnikoff et al. described an increase of the receptor CD66b in DED patients, a secondary marker of neutrophil degranulation [70]. Moreover, hyperosmolarity may promote neutrophil extracellular trap (NET) formation. NETs are defense structures formed after neutrophils release chromatin [71]. NETs can obstruct the ducts of meibomian glands, causing MGD [72]. As mentioned earlier, MGD is the main cause of EDE. 

Mast cells in the conjunctiva play pivotal roles in allergic conjunctivitis inflammation [73]. Upon activation, mast cells undergo degranulation and release mediators, such as histamine, tryptase, and chymase. Tryptase and chymase directly degrade extracellular matrix components [74]. Tryptase is a trypsin-like serine protease and is only present in mast cells, making it the perfect indicator of mast cell degranulation [75]. Li et al. studied the effect of tryptase on the corneal epithelial barrier function and described that human corneal epithelial cells (HCE) exposed to tryptase expresses MMP-9, and the corneal epithelial barrier is disturbed [76]. Similarly, Ebihara et al. described that chymase could also decrease the barrier function in HCE [77]. It is known that patients who suffer from ocular allergy can develop DED [78]. Their close relation derives the hypothesis that mast cell proteases might be present in patients with both conditions. 

Studies demonstrate the presence of immune cells on the ocular surface. However, further studies are needed to determine the presence of serine proteases released by neutrophils and mast cells in patients and validate their therapeutic potential for treating DED. 

### 2.3. Cysteine Proteases

Cysteine proteases, similar to serine proteases, are named after having cysteine as a nucleophile in their active site. They are divided into different clans based on sequence homology [79]. The most abundant is clan C1, which includes papain and calpain proteases [80]. Most cathepsins belong to this clan, excluding cathepsins A and G, which belong to the serine proteases, and cathepsin D and E, belonging to aspartic proteases. Cathepsins are lysosomal proteases, playing essential roles in the extracellular matrix. Cystatins regulate cathepsin activity. Their role is to mediate the release of proteases from lysosomes and protect the tissues from invading microorganisms or parasites [81]. 

Cathepsins play roles in arthritis, osteoporosis, Alzheimer’s disease, cancer, and apoptosis [82]. Focusing on the eye, cathepsins are found in the cornea, retinal pigment epithelial cells, optic nerve, and choroid [83]. However, their role in health and disease has not been investigated in depth.

Cathepsin S has been correlated with SSDE. It is known for protein and extracellular matrix degradation, thus facilitating cell migration and infiltration [84]. Cathepsin S is stored in lysosomes as a zymogen and activated in late endosomes or lysosomes [85]. Li et al. were the first to identify the upregulation of cathepsin S expression and activity in lacrimal gland acinar cells and subsequently in tears of NOD mice, a mice model of SSDE [86]. Later, Hamm-Alvarez et al. further studied the clinical potential of cathepsin S as a biomarker for SSDE. They found that the activity of cathepsin S is higher in SSDE patients compared with healthy controls or patients with NSSDE or other autoimmune diseases [87]. Cathepsin S may also affect the quality of tears by degradation of other tear proteins [88]. Yu et al. recently investigated the cathepsin S activity in age-related DED. Interestingly, they found a significant increase in cathepsin S expression and activity in aged C57BL/6J mice compared with young ones. Thus, an aged ocular surface might be prone to an increase in this protease. Although cathepsin S is not upregulated in all DED patients, it could be an interesting biomarker to discriminate between SSDE and NSSDE and also for age-related DED.

## 3. Protease-Activated Receptors and Dry Eye Disease

Proteases are also known for their role in activating PARs [89]. These belong to the seven-transmembrane G protein-coupled receptors (GPCRs). Thus, PARs are cell-surface proteins composed of seven transmembrane domains and three extracellular and three intracellular loops [90]. PARs differ from other GPCR receptors due to their activation by proteases instead of activation by a ligand. Especially, serine proteases can cleave their N-terminal exodomain by proteolysis. The new N-terminal sequence, acting as a tethered ligand, interacts with extracellular loop-2, leading to a conformational change of the receptor [91]. After that, intracellular signaling starts (Figure 2). However, this can vary from the receptor subtype, the protease responsible for activation, and the activated pathway. Contrarily, some proteases can inactivate PARs by cleaving the N-terminus in a different position [92]. Currently, four subtypes of PARs are described from PAR-1 to PAR-4 [89]. The functions and proteases that can cleave and activate them differ from subtypes [93]. This section focuses on PAR-2 since only studies on this receptor could be found with interest in the ocular surface. 

The canonical activation of PAR-2 can be performed by several trypsin-like serine proteases, including trypsin, thrombin, tryptase, matriptase, plasmin and some kallikreins. After activation, PAR-2 couples with G_αq/11_, which leads to the hydrolysis of phosphatidylinositol 4,5-bisphosphate (PIP_2_), and the Ca^2+^/inositol 1,4,5-triphosphate (IP_3_)/PKC signaling pathway is initiated [94]. This pathway leads to the activation of NFκB, which induces NFκB-dependent genes’ induction of proinflammatory cytokines and intracellular adhesion molecule-1 (ICAM-1) [95]. The canonical activation of PAR-2 has also been related to MAPK signaling and subsequent inflammatory response. Other proteases can activate PAR-2 in a biased manner, initiating different signaling pathways [96]. For instance, studies demonstrated that biased activation by neutrophil elastase activates MAPK [97]. The activation of MAPK and NFκB is also associated with DED inflammatory response [98].

PARs have been widely studied and demonstrated to be potential drug targets and valuable biomarkers for many pathologies. For instance, PAR-2 is correlated with visceral hypersensitivity in irritable bowel syndrome [99]; in the cardiovascular system, it is associated with vascular inflammation [100], and airway proteases and PARs have been proposed as therapeutic targets for various lung diseases [101].

While the presence of PARs has been widely studied in diverse human tissues and pathologies, not much is known about their expression and function in the different human eye tissues. The involvement of PARs in DED was never mentioned until recently when Joossen et al. described a significant increase in PAR-2 expression on the corneal tissue of untreated dry eye rats [38]. Several studies indicate the expression of PAR-1 and PAR-2 in HCE [76,102,103]. Lang et al. also demonstrated that a specific activation of PAR-2 by trypsin and thrombin increased the production of the proinflammatory cytokines IL-6, IL-8, and TNF-α by HCE [102]. These cytokines are also found on the tear film of DED patients. Li et al. postulated that tryptase activates PAR-2 in HCE, compromising the barrier function and triggering the expression of MMP-9 [76], another DED biomarker. Cathepsin S can activate PAR-2 in a distinctive site compared with the serine proteases, which proved to cause inflammation and neuropathic pain [104]. Klinngam et al. showed that the increased secretion of IL-6, IL-8, TNF-α, IL-1β, and MMP-9 in HCE treated with cathepsin S is correlated with PAR-2 expression [105].

Although little is known about the function of PAR-2 in DED, studies demonstrating their presence in corneal epithelial cells and their ability to increase the levels of proinflammatory cytokines support the hypothesis of the participation of these receptors in DED inflammatory responses.

## 4. Protease Inhibitors and Dry Eye Disease

Only a limited number of papers on this subject are reported. However, with evidence that proteases play a role in the pathophysiology of DED, protease inhibitors could potentially be new therapeutics. This section summarizes the studies and results with protease inhibitors in DED animal models (Table 1).

### 4.1. MMP-9 Inhibitors

Mori et al. studied PES_103, a synthetic water-soluble small molecule targeting the catalytic domain of MMP-9. They administered 0.1% PES_103 to an experimental reduced lacrimation mice model twice a day. This treatment showed an increase in tear production [106]. In another study, a divalent PAMAM-based MMP-9 inhibitor showed increased tear production in an experimental rabbit dry eye model, in which the administration of atropine sulfate by ocular instillations induced DED. Moreover, there was no sign of corneal damage [107]. Similarly, RSH-12, an MMP-9 peptide inhibitor, showed reduced signs of DED in a rabbit model. Treatment for 7 days showed an increase in tear volume and a decrease in tear breakup time [108].

### 4.2. Serine Protease Inhibitors

The serine protease inhibitor A3K (SERPINA3K) belongs to the endogenous serine protease inhibitor family. SERPINA3K has effects on other ocular treatments, such as corneal injury [109]. A study by Hu et al. described a reduced TNF-α-induced disruption of the rabbit corneal endothelial barrier [110]. Z. Lin et al. studied the effect of the inhibitor on dry eye and squamous metaplasia, an ocular surface pathology commonly occurring under prolonged tear deficiency conditions [111]. This study used a mouse animal model, where DED was induced by BAC [112]. Both BAC and SERPINA3K were administered twice a day for 16 days. Mice treated with SERPINA3K showed a significant reduction in DED development. They showed minor epithelial damage and reduced inflammatory response in the cornea. They postulate that SERPINA3K can decrease DED severity by reducing TNF-α. 

PEDF is a 50 kDa glycoprotein belonging to the serpin family [113]. In the eye, it is expressed in multiple tissues, such as the cornea, retina, choroid, and ciliary muscles [114]. Singh et al. first demonstrated that treatment with recombinant PEDF in a dry eye model inhibits the maturation of corneal dendritic cells, reduces Th17 generation, and suppresses the expression of proinflammatory cytokines on the ocular surface. The mice received 1 μL of 100 μL/mL topical recombinant murine PEDF via ocular surface instillations three times a day for 7 days. DED was induced in mice in a controlled environment chamber [115]. A subsequent study postulates that PEDF reduces DED severity by significantly reducing corneal fluorescein staining scores [116].

Recently, Joossen et al. described a serine protease inhibitor, UAMC-00050, to treat inflammation in DED [38]. UAMC-00050 is a synthetic diaryl phosphonate small-molecule inhibitor that targets several trypsin-like serine proteases [117]. They reported in vivo studies carried out in a rat animal model, where DED was induced by the surgical removal of the exorbital lacrimal gland [118]. The animals were treated for 24 days, twice a day. UAMC-00050 at a concentration of 5 mM appeared to be the most promising treatment compared with cyclosporin A and vehicle animals. Fluorescein scores showed that UAMC-00050 reduces tissue damage significantly, and the expression of inflammatory cytokines, IL-1α and TFN-α, were significantly reduced. UAMC-00050, at a concentration of 5 mM, was able to reduce inflammatory cell infiltration (CD3^+^ and CD45^+^) and MMP-9 activity. The accumulation of pro-MMP9 and the decrease in active MMP-9 could show that serine proteases have a role in activating MMP-9 in DED [38].

## 5. Conclusions and Future Directions/Perspectives

Tear proteins are a good indicator of health and disease on the ocular surface. The metalloprotease MMP-9 has been widely studied as a biomarker of DED, and it is commonly used in clinical settings to diagnose DED. There is enough evidence that not only MMP-9 is a relevant protease in DED pathophysiology. Furthermore, protease inhibitors could be used as a potential treatment for DED, which would tackle inflammation directly. However, there is a lack of studies focused on different protease families, such as the serine or cysteine proteases. Even though the tear proteome is very complex, identifying upregulated proteases in DED patients could open a window for new biomarkers and potentially new protease inhibitors as a novel DED treatment.

Several techniques can monitor and quantify protein expression in complex proteomes, for example, the enzyme-linked immunosorbent assay (ELISA) or protein microarrays [120,121]. Other techniques, such as isotope-coded affinity tag labelling (ICAT), stable isotope labelling by amino acids in cell culture (SILAC), and isobaric tag for relative and absolute quantitation (iTRAQ), were developed for quantitative proteomics [122,123,124]. However, these techniques analyze protein expression rather than activity. The activity of proteins is regulated by post-translational modifications, such as zymogen activation or interaction with other proteins or small molecules [123]. Therefore, it is of interest to identify active proteases. Activity-based protein profiling (ABPP) is a proteomic tool that uses activity-based probes (ABPs) to identify active enzymes in a complex proteome. The reactive group of the probe interacts with the active site of the protease, and later, it is identified by electrophoresis or mass spectrometry (Figure 3) [125]. Peng et al. published an interesting review on the prospective application of ABPP with ocular proteases [126].

After identifying upregulated proteases in DED, efforts could be moved to the design of selective inhibitors for a specific protease with therapeutic potential. Other techniques, such as X-ray crystallography, can determine the protease structure and, together with molecular docking, help to design novel inhibitors [121,127].

## Figures and Tables

**Figure 1 ijms-23-09795-f001:**
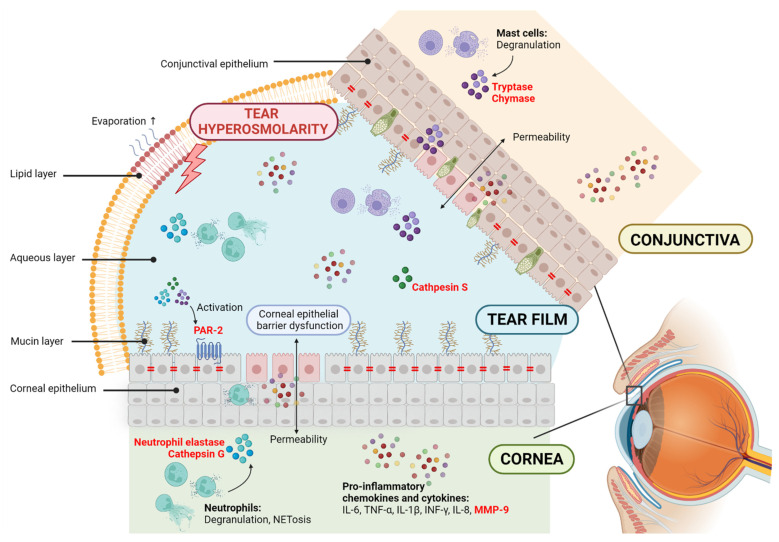
Potential contribution of proteases in dry eye disease (DED). The disruption of homeostasis of the tear film is accompanied by tear instability and hyperosmolarity and ocular surface inflammation and damage. DED is related to corneal epithelial barrier dysfunction, allowing permeability and cell circulation to the tear film. This elevates the production of proinflammatory chemokines and cytokines, including the metalloprotease MMP-9. Neutrophils and mast cells are innate immune cells in the cornea and conjunctiva that, upon degranulation, release biological mediators to the environment, including serine proteases. Cathepsin S is a cysteine protease found in the tears of Sjögren syndrome patients. Proteases are known to promote the expression and activation of proinflammatory cytokines and impact the degradation of extracellular matrix components and the loss of epithelial barrier function. Proteases are also known for activating protease-activated receptors (PARs) and starting intracellular signaling. PAR-2 is expressed in corneal epithelial cells. In red, the proteases and protease-activated receptors are highlighted.

**Figure 2 ijms-23-09795-f002:**
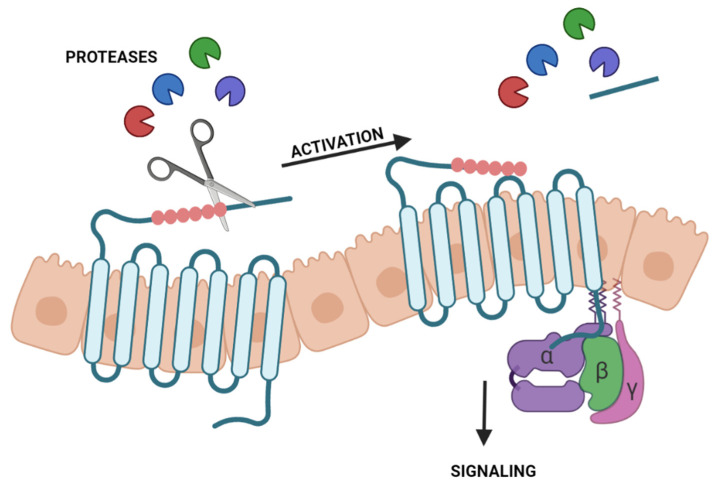
Activation of protease-activated receptors (PARs) by proteases. Proteases cleave the N-terminal exodomain of a specific PAR. The new N-terminal acting as a tethered ligand interacts with the extracellular loop-2 starting intracellular signaling.

**Figure 3 ijms-23-09795-f003:**
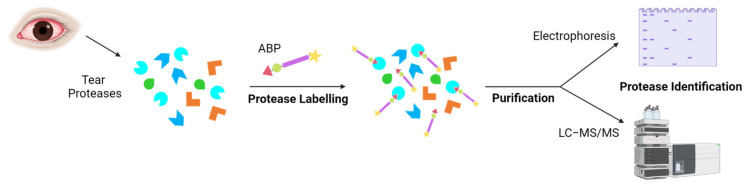
Tear protease identification by activity-based protein probes (ABPs). Proteases present in DED patients are labelled with an ABP for a specific protease family. The reactive group of the ABP would bind to the active enzymes. The bound proteases can be isolated and analyzed by electrophoresis and/or mass spectrometry.

**Table 1 ijms-23-09795-t001:** Protease inhibitors and the effect they have in a specific experimental setting.

Inhibitor	Target	Experimental Setting	Effect ^a^	Ref.
PES_103	MMP-9	Dry eye mice model Transdermal scopolamine patches	↑ Tear production	[106]
Divalent PAMAM	MMP-9	Dry eye rabbit model Atropine sulfate	↑ Tear production ↓ Corneal damage	[107]
RSH-12	MMP-9	Dry eye rabbit model Atropine sulfate	↑ Tear volume ↓ Tear breakup time	[108]
SERPINA3K	Serine proteases	Dry eye mice model BAC induced	↓ Epithelial damage ↓ TNF-α	[111]
PEDF	Serine protease	Dry eye mice model Controlled environment chamber	↓ DCs, Th17 ↓ Proinflammatory cytokines ↓ Fluorescein score	[115,119]
UAMC-00050	Serine proteases	Dry eye rat model Surgical removal exorbital lacrimal gland	↓ IL-1α, TNF-α, MMP-9 ↓ CD3+, CD45+	[38]

^a^ An up-facing arrow (↑) represents an increase, whereas a down-facing arrow (↓) corresponds to a reduction.

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
