# Peer review of "Proteases and Their Potential Role as Biomarkers and Drug Targets in Dry Eye Disease and Ocular Surface Dysfunction"

_ijms, 2022, doi:10.3390/ijms23179795_

Round 1
Reviewer 1 Report
This manuscript reviewed the implications of seven families of proteases and their inhibitors in the pathogenesis of dye eye disease (DED). This is much needed review that summarized current research on protease as biomarkers and potential therapeutic targets for DED. This review is well written with sufficient references cited. It will promote further investigation on proteases for new biomarkers and potential treatment of DED. I only have a suggestion if this review includes the role of MMPs and other proteases in activating pro-inflammatory cytokines from their inactive precursors or proforms.
Author Response
Dear reviewer,
We would like to thank you for your comments to improve our manuscript. We appreciate the suggestion of adding the role of MMPs and proteases in activating the inactive forms of pro-inflammatory cytokines; however, we believe that the manuscript already states the role of proteases in activating the expression of pro-inflammatory cytokines. For example, in line 125: “Proteases are known to promote the expression of pro-inflammatory cytokines…”. Nevertheless, we recognize this should be mentioned in the paper, so we added the following: “Proteases are known to promote the expression and activation of pro-inflammatory cytokines…”.
Reviewer 2 Report
The manuscript titled “Proteases and Their Potential Role as Biomarkers and Drug Targets in Dry Eye Disease and Ocular Surface Dysfunction” by Ramos-Llorca, A.; et al. is a review where the authors highlight the action of proteases with different nature on dry eye diseases (DED) and how they could act as suitable biomarkers to design the next-generation of therapies against this type of human disorders. The present work can significantly aid to better understand the human immunity response against proteases, how to detect and identify them and finally, the selection optimization of protease inhibitors to develop efficient treatments. Moreover, this study is highly complementary to other scientific works existing in literature which could aid to increase the positive impact on society (e.g. other kind of proteases with alternative functions respect to DED). This work is relevant not only based on the above described reasons, but also because it may be fully extendable for human viral diseases that involve protease functions (e.g. hepatitis, VIH, human papillomavirus, among others). The gathered findings may be relevant for the examined field. The results achieved are well-discussed during the main body of the reported manuscript. The scientific paper is well written. In my opinion the present manuscript is innovative and the methodological approached used matches with the scope of International Journal of Molecular Sciences. For the above described reasons, I recommend the publication in International Journal of Molecular Sciences once the following remarks will be fixed:
--------
INTRODUCTION
The information and references detailed in the Introduction section are accurate regarding the content of the submitted manuscript. Nevertheless, authors should take care of the following minor remarks:
I) Authors should define the abbreviation “TFOS DEWS II report” (line 22) by adding the following statement: “Tear Film & Ocular Society (TFOS) launched the Dry Eye Workshop II (DEWS) report”.
II) “Risk factors for DED include sex and race” (lines 32-33). Please, the authors should incorporate some bibliography citation to make this affirmation [1].
[1] Ahn, J.H.; et al. Sex differences in the effect of aging on dry eye disease. Clin. Interv. Aging 2017, 12, 1331-1338. https://doi.org/10.2147/CIA.S140912.
III) “(…) FDA-approved drug (…)” (line 70). Please, define the term FDA. The aforementioned sentence should be replaced by “(…) Food and Drug Administration (FDA, USA) (…)”.
--------
PROTEASES AND DRY EYE DISEASE
I) “Proteases play pivotal roles (…). Different reviews can be found on this subject [27, 38].” (lines 107-109). Authors have introduced two relevant reviews in this field. Nevertheless, maybe it may be convenient to add one extra review work to have a more complete overview of the importance of proteases as targets on therapy treatments [2].
[2] Bleuez, C.; et al. Exploiting protease activation for theraphy. Drug Discov. Today 2022, 27, 1743-1754. https://doi.org/10.1016/j.drudis.2022.03.011.
II) “We summarize (…) (Error! Reference source not found).” (lines 112-114). This issue should be fixed. Authors should take care and check the source of the above described problem.
III) Figure 1 caption (lines 116-126). Please, authors should describe the meaning of red bold terms in the Figure caption.
--------
PROTEASE-ACTIVATED RECEPTORS AND DRY EYE DISEASE
“After that, intracellular signalling starts (Error! Reference source not found.)” (line 238). This issue should be fixed. Authors should take care and check the source of this bibliography citation. Similar warning advices appear on lines 287 and 345.
--------
CONCLUSIONS AND FUTURE DIRECTION
Authors should consider to change the title of this section by “Conclusions and future directions/perspectives”. Then, authors listed electrophoresis and high-performance liquid chromatography (HPLC)/mass spectrometry (MS) as potential techniques to decipher the protease nature identification and its natural inhibitors. Authors should include some further techniques on this regard to make this section more robust. In this framework, ELISA immunodetection with specific antibodies is also presented as powerful toolbox to identify proteases and its inhibitors [3]. Moreover, atomic force microscopy [4] has successfully devoted to address protease morphologies upon ligand and catalysis conditions [5]. Finally, X-ray crystallography can determine the protease structure in vitro conditions [6] or in combination with molecular docking experiments [7]. All aforementioned techniques could eventually be employed for those proteases involved in DED infections.
[3] Saarinen, N.V.V.; et al. Antibody Responses against Enterovirus Proteases are Potential Markers for an Acute Infection. Viruses 2020, 12, 78. https://doi.org/10.3390/v12010078..
[4] Marcuello, C.; et al. Atomic Force Microscopy to Elicit Conformational Transitions of Ferredoxin-Dependent Flavin Thioredoxin Reductases. Antioxidants 2021, 10, 1437. https://doi.org/10.3390/antiox10091437.
[5] Vega, S.; et al.; NS3 protease from hepatitis C virus: biophysical studies on intrinsically disordered protein domain. Int. J. Mol. Sci. 2013, 14, 13282-13306. https://doi.org/10.3390/ijms140713282.
[6] Zhang, L.; et al. Crystal structure of SARS-CoV-2 main protease provides a basis for design of improved α-ketoamide inhibitors. Science 2020, 368, 409-412. https://doi.org/10.1126/science.abb3405.
[7] Costanzi, E.; et al. Structural and Biochemical Analysis of the Dual Inhibition of MG-132 against SARS-CoV-2 Main Protease (Mpro/3CLpro) and Human Cathepsin-L. Int. J. Mol. Sci. 2021, 22, 11779. https://doi.org/10.3390/ijms222111779.
--------
REFERENCES
All bibliography citations are in the proper format of International Journal of Molecular Sciences.
--------
OVERVIEW AND FINAL COMMENTS
The submitted work is well-designed and the gathered results are interesting for the biomedical and clinical fields in special related for drug screening and pharmacokinetics applications. For this reason, I will recommend the present scientific manuscript for further publication in International Journal of Molecular Sciences once all the aforementioned suggestions will be properly fixed.
Author Response
Dear reviewer,
We would like to thank you for your thoughtful comments and suggestions to improve our manuscript. We address your recommendations here.
_____
INTRODUCTION
Comment I & III: Authors should define the abbreviation “TFOS DEWS II report” (line 22) by adding the following statement: “Tear Film & Ocular Society (TFOS) launched the Dry Eye Workshop II (DEWS) report”; “(…) FDA-approved drug (…)” (line 70). Please, define the term FDA. The aforementioned sentence should be replaced by “(…) Food and Drug Administration (FDA, USA) (…)”.
The abbreviations have been described in the main text and the abbreviation list.
Comment II: “Risk factors for DED include sex and race” (lines 32-33). Please, the authors should incorporate some bibliography citation to make this affirmation [1].
[1] Ahn, J.H.; et al. Sex differences in the effect of aging on dry eye disease. Clin. Interv. Aging 2017, 12, 1331-1338. https://doi.org/10.2147/CIA.S140912.
The recommended reference has been added.
____
PROTEASES AND DRY EYE DISEASE
Comment I: “Proteases play pivotal roles (…). Different reviews can be found on this subject [27, 38].” (lines 107-109). Authors have introduced two relevant reviews in this field. Nevertheless, maybe it may be convenient to add one extra review work to have a more complete overview of the importance of proteases as targets on therapy treatments [2].
[2] Bleuez, C.; et al. Exploiting protease activation for theraphy. Drug Discov. Today 2022, 27, 1743-1754. https://doi.org/10.1016/j.drudis.2022.03.011.
A third reference with an additional review has been added.
Comment II: “We summarize (…) (Error! Reference source not found).” (lines 112-114). This issue should be fixed. Authors should take care and check the source of the above described problem.
The cross-references have been corrected. This also applies to the comment in section PROTEASE-ACTIVATED RECEPTORS AND DRY EYE DISEASE.
Comment III: Figure 1 caption (lines 116-126). Please, authors should describe the meaning of red bold terms in the Figure caption.
The following sentence has been added to describe the red bold terms in Figure 1: “In red, the proteases and protease-activated receptors are highlighted.”
_____
CONCLUSIONS AND FUTURE DIRECTION
Authors should consider to change the title of this section by “Conclusions and future directions/perspectives”. Then, authors listed electrophoresis and high-performance liquid chromatography (HPLC)/mass spectrometry (MS) as potential techniques to decipher the protease nature identification and its natural inhibitors. Authors should include some further techniques on this regard to make this section more robust. In this framework, ELISA immunodetection with specific antibodies is also presented as powerful toolbox to identify proteases and its inhibitors [3]. Moreover, atomic force microscopy [4] has successfully devoted to address protease morphologies upon ligand and catalysis conditions [5]. Finally, X-ray crystallography can determine the protease structure in vitro conditions [6] or in combination with molecular docking experiments [7]. All aforementioned techniques could eventually be employed for those proteases involved in DED infections.
[3] Saarinen, N.V.V.; et al. Antibody Responses against Enterovirus Proteases are Potential Markers for an Acute Infection. Viruses 2020, 12, 78. https://doi.org/10.3390/v12010078..
[4] Marcuello, C.; et al. Atomic Force Microscopy to Elicit Conformational Transitions of Ferredoxin-Dependent Flavin Thioredoxin Reductases. Antioxidants 2021, 10, 1437. https://doi.org/10.3390/antiox10091437.
[5] Vega, S.; et al.; NS3 protease from hepatitis C virus: biophysical studies on intrinsically disordered protein domain. Int. J. Mol. Sci. 2013, 14, 13282-13306. https://doi.org/10.3390/ijms140713282.
[6] Zhang, L.; et al. Crystal structure of SARS-CoV-2 main protease provides a basis for design of improved α-ketoamide inhibitors. Science 2020, 368, 409-412. https://doi.org/10.1126/science.abb3405.
[7] Costanzi, E.; et al. Structural and Biochemical Analysis of the Dual Inhibition of MG-132 against SARS-CoV-2 Main Protease (Mpro/3CLpro) and Human Cathepsin-L. Int. J. Mol. Sci. 2021, 22, 11779. https://doi.org/10.3390/ijms222111779.
The title has been modified to “Conclusions and future directions/perspectives”
Further techniques to monitor protease expression have been added. These include ELISA, protein microarrays, and other proteomic techniques such as ICAT, SILAC or iTRAQ. Moreover, techniques such as X-ray crystallography and molecular docking have been added to help determine the protease structure and design novel inhibitors. All abbreviations have been added to the Abbreviation list.
Reviewer 3 Report
I very much welcome the publication of this interesting article.
I only found the following minor recommendations:
Introduction
Lines 23-24: I recommend to replace ‘symptoms’ by ‘signs’, as the symptoms are described in the sentence after.
Lines 59 – 60: should read ‘assesses’ instead of ‘assess’.
Author Response
Dear reviewer,
We would like to thank you for your comments and suggestions to improve our manuscript. We address your recommendations here.
Lines 23-24: I recommend to replace ‘symptoms’ by ‘signs’, as the symptoms are described in the sentence after.
The word symptoms has been replaced by signs.
Lines 59 – 60: should read ‘assesses’ instead of ‘assess’.
The word assess has been modified to assesses.